# LECO: Learnable Episodic Count for Task-Specific Intrinsic Reward

**Daejin Jo***      **Sungwoong Kim***      **Daniel Wontae Nam***

**Taehwan Kwon**      **Seungeun Rho**      **Jongmin Kim**      **Donghoon Lee**

Kakao Brain
Seongnam, South Korea
{daejin.jo, swkim, dwtnam, taehwan.kwon, seungeun.rho, jmkim, dhlee}
@kakaobrain.com

## Abstract

Episodic count has been widely used to design a simple yet effective intrinsic motivation for reinforcement learning with a sparse reward. However, the use of episodic count in a high-dimensional state space as well as over a long episode time requires a thorough state compression and fast hashing, which hinders rigorous exploitation of it in such hard and complex exploration environments. Moreover, the interference from task-irrelevant observations in the episodic count may cause its intrinsic motivation to overlook task-related important changes of states, and the novelty in an episodic manner can lead to repeatedly revisit the familiar states across episodes. In order to resolve these issues, in this paper, we propose a learnable hash-based episodic count, which we name LECO, that efficiently performs as a task-specific intrinsic reward in hard exploration problems. In particular, the proposed intrinsic reward consists of the episodic novelty and the task-specific modulation where the former employs a vector quantized variational autoencoder to automatically obtain the discrete state codes for fast counting while the latter regulates the episodic novelty by learning a modulator to optimize the task-specific extrinsic reward. The proposed LECO specifically enables the automatic transition from exploration to exploitation during reinforcement learning. We experimentally show that in contrast to the previous exploration methods LECO successfully solves hard exploration problems and also scales to large state spaces through the most difficult tasks in MiniGrid and DMLab environments.

## 1 Introduction

In reinforcement learning (RL), tackling the problem of sparse rewards still remains a great challenge. Many works have been proposed to develop dense synthetic reward, which is generally called intrinsic reward, to provide the motivation for exploration in the absence of extrinsic rewards from the environment [5, 26, 1, 28, 13, 38, 33, 10, 40, 29, 37, 15, 21]. Among them, count-based exploration [31, 4, 25, 24, 33, 23, 5], that encourages the RL agent to visit novel states, is one of the most popular classes for intrinsic motivation since it is simple and effective in many tasks, and more importantly it is theoretically justified [31, 20]. In general, the state novelty can be measured within a single episode or through an entire task across episodes. In recent RL problems, environments are often changed over episodes or even procedurally generated within the same task [7, 3, 22, 8, 17]. Therefore,

---

*Contributed equally.

36th Conference on Neural Information Processing Systems (NeurIPS 2022).

per-episode novelty by *episodic count* has been widely adapted in modern hard exploration problems [1, 28, 38].

However, in high-dimensional and/or continuous state spaces, state counts need to be approximated in an abstract space since most states are likely to be unique and novel in the original space, leading to insufficient exploration [4, 25, 28, 1]. In addition, computation of episodic count at every time step based on the episodic memory could be burdensome if an episode lasts for a long time and therefore requires fast hashing [33]. Previous works have discretized the state space to obtain the hash code [33, 14, 9] for a fast pseudo-count, but they have not extensively applied it to recent difficult exploration problems such as procedurally generated and partially observable environments.

Moreover, existing count-based exploration methods have mostly employed task-agnostic state novelty, and thereby task-irrelevant count may disturb its intrinsic reward bonus to faithfully reflect task-related information [19, 2, 18, 40]. This problem could be severe in exploration by per-episode count. For example, task-wise familiar states across episodes can have high novelty in each episode, which confines overall exploration into smaller areas, while some states that are highly related to extrinsic rewards need to be repeatedly revisited even in a single episode for obtaining extrinsic rewards more densely.

In this work, we propose a hash-based episodic count that is learned to provide task-specific intrinsic motivation in solving hard exploration problems. The proposed **L**earnable **E**pisodic **CO**unt (LECO) produces task-specific intrinsic rewards that are efficiently and practically applicable to high-dimensional states, long-horizon episodes, procedurally generated environments, and highly sparse extrinsic rewards. In specific, the proposed intrinsic reward is composed of the episodic count-based novelty and the task-specific modulation. Our episodic count makes use of a vector quantized variational autoencoder (VQ-VAE) [34] to automatically obtain the discrete state codes for a fast counting. In addition, we exploit a learnable modulator to control the episodic novelty in relation to task-specific extrinsic rewards. The proposed LECO also adaptively shifts the RL phase from exploration to exploitation. Experimental results on the most difficult exploration tasks in procedurally generated environments of MiniGrid [7] and DMLab [3] show that LECO efficiently solves such tasks and outperforms previous state-of-the-art exploration methods.

Our main contributions can be summarized as follows:

- A novel learnable episodic count is proposed to efficiently provide task-specific intrinsic motivation to the RL agent under complex environments with sparse rewards.
- A complementary synthesis of learnable hash code based on VQ-VAE and learnable intrinsic reward modulator enables task-specific transition from exploration to exploitation.
- The proposed LECO demonstrates consistent performance improvements over previous exploration methods in solving various hard exploration benchmark tasks.

## 2   Related Work

**Count-based exploration.**   Count-based exploration was first proposed in tabular tasks [31], which promotes the agent to reduce its uncertainty by visiting unfamiliar states corresponding to low counts. However, it is basically ineffective in high-dimensional state spaces since every state would be different. Therefore, several recent works have utilized count-based exploration for high-dimensional environments. For instance, pseudo counts are derived from density models using context tree switching [4] or PixelCNN [25]. More recently, RND [5] has approximated the state visitation count by the difference between a random fixed target network and a trained predictor network. RND has also been used in other methods for measuring the state novelty [1, 38, 37]. On the other hand, a number of algorithms have transformed the high-dimensional state to the discretized hash code for fast visitation counting by a hash table [33, 10]. Similarly, we obtain the episodic count by the discretized hash code. However, we exploit learnable hash codes by VQ-VAE that is trained to approximate the state density. In addition, different from [14] that directly produces a global code by VQ-VAE, we concatenate local codes as in the original VQ-VAE for generating a single state code.

**Task-specific intrinsic motivation.**   While count-based exploration methods have typically utilized the state novelty in a task-agnostic manner, curiosity-based methods have tried to consider task-related information for intrinsic motivation by learning the environment dynamics [26, 29, 19, 28]. They

have developed intrinsic rewards based on the prediction error upon the agent's actions. Among them, recently, a number of approaches have exploited the task-dependent state novelty using the information bottleneck in order to ignore task-irrelevant distractions [19, 2, 18]. Meanwhile, there have been a number of attempts to directly learn the intrinsic reward function to ultimately optimize the task-specific extrinsic rewards [40, 39, 36]. Their algorithms are generally performed under the meta-gradient framework. LECO also adopts the idea of learnable rewards and the gradient-based bi-level optimization [40] to construct a task-specific modulator for intrinsic rewards. However, unlike those works that do not work well with sparse extrinsic rewards, LECO takes a full advantage of intrinsic motivation by the task-agnostic state novelty based on the episodic count before obtaining the extrinsic rewards.

**Exploration for procedurally generated environments.**   One of the important challenging RL problems is robust exploration in a procedurally generated environment where the environment (partially) changes between episodes, and hence the agent needs to explore a very large state space and have strong generalization ability. Many recent RL benchmarks deal with procedurally generated environments [7, 3, 22, 8, 17], and accordingly state-of-the-art exploration algorithms have tried to improve sample efficiency in such environments. RIDE [28] has used a change between successive states on a latent space as an intrinsic reward while AGAC [13] has encouraged action diversity by an adversarial policy. NovelD [38] has exploited regulated difference of the novelty between consecutive states, and MADE [37] has devised exploration by maximizing the deviation from previously explored regions. AMIGo [6] has proposed a teacher that generates count-based intrinsic goals. In contrast to these handcrafted or task-agnostic intrinsic rewards, LECO learns to leverage task-level information to guide the count-based exploration especially at the episode level due to the procedurally generated environment. Moreover, LECO also shows improved performances on DMLab tasks with higher-dimensional partial observations.

## 3   Learnable Episodic Count via Task-Specific Modulation

### 3.1   Notation

Our problem setting follows a typical RL problem described using Markov Decision Process defined by a set of states $\mathcal{S}$, a set of actions $\mathcal{A}$, and a transition function $\mathcal{T} : \mathcal{S} \times \mathcal{A} \rightarrow \mathcal{S}$. An RL agent interacting with the environment samples its actions from a policy, which is an action distribution function $\pi \rightarrow \mathcal{P}(\mathcal{A})$, and receives a reward, $r : \mathcal{S} \times \mathcal{A} \rightarrow \mathbb{R}$, upon state transition. The objective of the agent is to learn a policy that maximizes the expected sum of rewards $G_t = \mathbb{E}_\pi \left[ \sum_{k=0}^{\infty} \gamma^k r_{t+k+1} \right]$, where $\gamma \in [0, 1]$ is a discount factor.

Among various sub-problems that RL faces, balancing the trade-off between exploration and exploitation still remains a fundamental problem [32]. Especially, efficient exploration is important when dealing with sparse rewards. One approach to solving such problem is to design an algorithm that directly generates exploring behavior. More specifically, a widely adopted method is to design an intrinsic motivation through a form of a reward such that the reward which the agent receives at time $t$ is given by $r_t = r_t^e + \alpha r_t^i$, where $r^e$ is the extrinsic reward from the environment, $r^i$ is the intrinsic reward designed for exploration, and $\alpha$ is a hyperparameter for scaling [28].

### 3.2   LECO Intrinsic Reward

The use of state visitation count for intrinsic rewards based on the state novelty has been a core concept in several previous studies [28, 1, 38, 13, 30, 16] due to its simplicity, task-agnostic property, and theoretical basis. However, such state-novelty measured from visitation counts monotonically decreases as more experiences accumulate, equally for both important and unimportant states. Thus, an agent would be discouraged to visit even the task-important states after a certain period of time, and the exploration towards those states vanishes [1]. Furthermore, an intrinsic bonus based on the state visitation could lead the agent to only pursue any state novelty, especially in the partially-observable and sparse reward environments. For example, imagine a task in which the goal is to find a colored key hidden in a box and unlock a door that has the same color as the key. If the box changes its color to a random color every time when it is opened, the agent would repeatedly open and close the box just to obtain the intrinsic bonus. In this task, the agent should be encouraged to find the door after obtaining the key and discouraged to repeat task-irrelevant actions. We directly build upon this idea

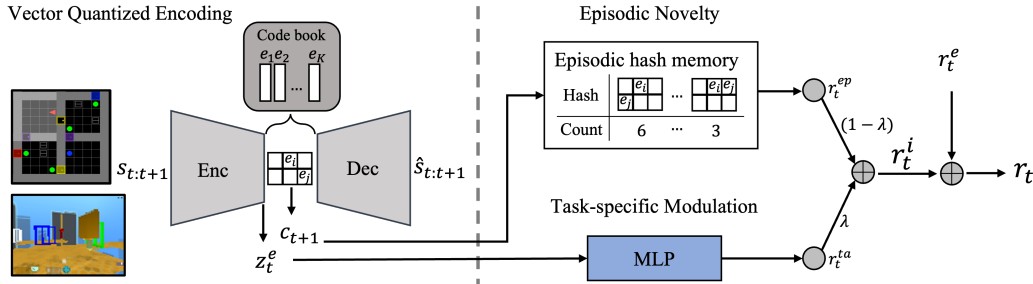

Figure 1: The schematic diagram of the intrinsic reward in LECO. The states at time $t$ and $t + 1$ are encoded and quantized to the embedding vector $z_t^e$ and hash $c_{t+1}$. Then, each is fed into the task-specific modulator and episodic counter, respectively, to produce the task-specific modulation $r_t^{\text{ta}}$ and episodic intrinsic reward $r_t^{\text{ep}}$, which are then combined to produce LECO's intrinsic reward $r_t^i$.

by decomposing the intrinsic reward into the state-novelty and task-specific modulation, to balance exploration and exploitation, respectively.

We design our intrinsic motivation to encourage the discovery of state novelty that is adaptively controlled to maximize the extrinsic return, hence the name **L**earnable **E**pisodic **CO**unt for task-specific motivation. Specifically, we define the general form of intrinsic reward as a sum of two parts,

$$r_t^i(a_{t-1}, s_t, a_t, s_{t+1}) = (1 - \lambda)r_t^{\text{ep}}(s_{t+1}) + \lambda r_t^{\text{ta}}(a_{t-1}, s_t, a_t), \qquad (1)$$

where $r_t^{\text{ep}}(s_{t+1}) = 1/\sqrt{N_{\text{ep}}(s_{t+1})}$ is the intrinsic reward from the episodic state-novelty, $N_{\text{ep}}(s)$ is the episodic visitation count of state $s$, and $r^{\text{ta}}$ is the task-specific modulation with $\lambda \in [0, 1]$ as its weighting factor. The first term measures the episodic novelty by state visitation count within the current episode that is initialized to zero at the beginning of each episode. The second term learns to modulate the episodic novelty in the first term according to whether the state is beneficial or harmful in achieving the task-specific objective. Note that the previous action $a_{t-1}$ is included as the input to reward calculation to provide extra transition information. A schematic diagram of the proposed intrinsic reward can be found in Figure 1.

LECO comprises of two submodules: 1) a learnable hash that encodes input observations to discrete hash codes for episodic counts and 2) a modulator that maximizes the task-specific extrinsic return by adaptively scaling the episodic novelty. Here, the modulator additively scales the episodic novelty. Compared to multiplicative scaling, the additive modulation allows the episodic novelty to be retained solely when the modulator does not generate enough signals due to the scarcity of extrinsic rewards.

### 3.3 Episodic Count via Vector Quantized Hashing

Previous researches have extended classic state visitation counts [31] to high-dimensional state spaces [4, 33]. A generalization of the classic count-based approach by locality-sensitive hashing (LSH) on the continuous embedding space was proposed [33] where the latent space is obtained by an autoencoder (AE). The use of a hash table in their approach enables counting the occurrences in a constant time, taking the advantage of scalability, especially in modern distributed RL systems [12, 11, 27]. However, such LSH does not take into account the spatial information due to its data independency [35]. As an alternative to AE, we propose to utilize vector quantized variational autoencoder (VQ-VAE) [34] that directly optimizes the discrete compressed representation of a state, producing a hash code without LSH.

In LECO, a state $s_t$ is mapped to a hash code $c_t$ by vector quantized hashing based on VQ-VAE comprised of an encoder, a decoder, and a codebook with $\theta_{\text{enc}}$, $\theta_{\text{dec}}$, and $\theta_e = \{e_k \in \mathbb{R}^D\}_{k=1}^K$ as their respective parameters. Here, $K$ is the number of codes, and $e_k$ is the $D$-dimensional embedding of the $k$th code. In detail, first, the input state $s_t$ is encoded to a spatial representation $z_t^e = f_{\text{enc}}(s_t; \theta_{\text{enc}})$, where $z_t^e \in \mathbb{R}^{w \times h \times D}$, and $f_{\text{enc}}(\cdot; \theta_{\text{enc}})$ is an encoder of VQ-VAE. Then, the representation $z_t^e$ is mapped to $z_t^q$ by finding the nearest vector in the codebook $\theta_e$ for each vector in the spatial position $i \in \{1, 2, ..., w \times h\}$. To be more specific, let us denote the vector in position $i$ of $z_t^e$ and $z_t^q$ as $z_t^e(i)$ and $z_t^q(i)$, respectively. Then, $z_t^q(i) = \arg\min_{e_k \in \theta_e} \|z_t^e(i) - e_k\|_2$, and the hash code is defined as

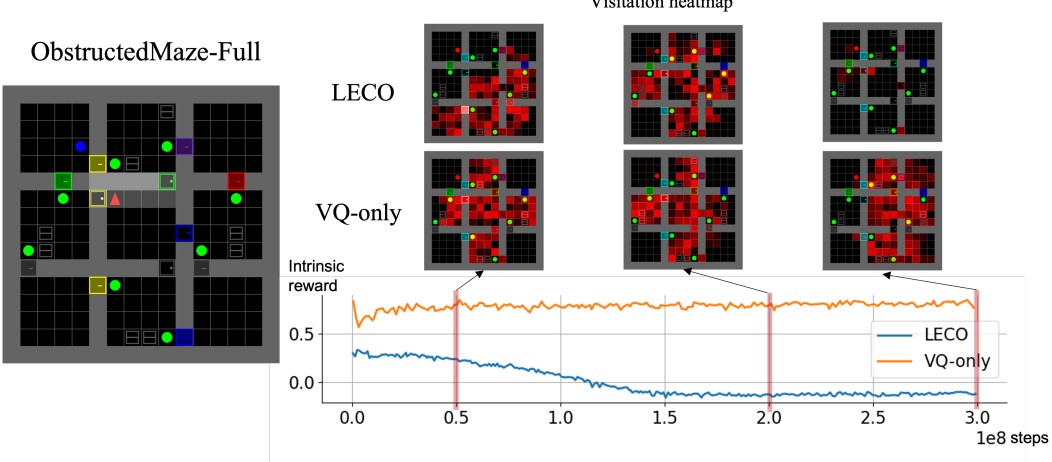

Figure 2: `ObstructedMaze-Full` Environment (**Left**). The changes in the coverage of visitations (**Right Top**) as the training progresses show that LECO modulates the episodic reward (**Right Bottom**) and so the agent focuses on the task-specific objective. On the other hand, the agent with intrinsic motivation from the episodic count only, labeled as VQ-only, remains distracted by the episodic state-novelty even in the late training phase. We provide more examples with further discussion in subsection D.2 in Appendix.

the sequence of indices such that $c_t = [k_i]_{i=1}^{w \times h}$ where $k_i = \arg\min_k \|z_t^e(i) - e_k\|_2$. Finally, the decoder then predicts the mean $\mu = f_{\text{dec}}(z_t^q; \theta_{\text{dec}})$ of an isotropic Gaussian distribution $p(s|z_t^q)$ with $\sigma^2 = 1$, where $f_{\text{dec}}(\cdot; \theta_{\text{dec}})$ is the decoder of the VQ-VAE.

Following the conventional VQ-VAE training, the hash function is updated according to the gradient

$$\nabla_{\theta_{\text{vq}}} \mathcal{J}^{\text{vq}} = -\sum_{t=0}^{T} \nabla \left( \log p(s_t|z_t^q) + \|\mathbf{sg}[z_t^e] - z_t^q\|_2^2 + \|z_t^e - \mathbf{sg}[z_t^q]\|_2^2 \right), \tag{2}$$

where $\mathbf{sg}$ stands for stopgradient operator, $T$ is an unroll length of the agent, and $\theta_{\text{vq}} = \{\theta_{\text{enc}}, \theta_{\text{dec}}, \theta_e\}$.

### 3.4 Task-Specific Intrinsic Modulation

In a sparse reward environment, when an extrinsic reward signal has little or no presence, having a task-agnostic intrinsic motivation can be significantly beneficial in guiding the agent to efficiently explore the environment until the discovery of more extrinsic signals. However, a task-general intrinsic reward such as the inverse of the episodic count may become a distracting signal in the later exploitation since it is not reduced with the progression of learning. We thereby propose to include task-specific modulation that aims to ultimately maximize the extrinsic return in the intrinsic reward formulation of LECO. The task-specific modulation can be seen as a dynamic factor for controlling the exploration and exploitation, based on the extrinsic rewards earned from the environment.

For this, we leverage the bi-level optimization proposed in LIRPG [40]. First, let policy $\pi$ parameterized by $\theta$ be updated using the policy gradient on the sum of intrinsic and extrinsic rewards,

$$\theta' = \theta + \eta \nabla_\theta \mathcal{J}^{e+i} = \theta + \eta \nabla_\theta \left( \mathbb{E}_{\pi_\theta}[\sum_{t=0}^{\infty} \gamma^t (r_t^e + \alpha r_t^i)] \right), \tag{3}$$

where $\eta$ is the step size. We then construct our task-specific modulator for the intrinsic reward in Equation 1 as a parametric model such that $r_t^i = (1 - \lambda) r_t^{\text{ep}} + \lambda r_t^{\text{ta}}(a_{t-1}, s_t, a_t; \theta_{\text{ta}})$ where $\theta_{\text{ta}}$ is the modulator parameters. Now, the task-specific modulator is trained to maximize only the extrinsic rewards using the corresponding policy gradient such that

$$\theta'_{\text{ta}} = \theta_{\text{ta}} + \eta_{\text{ta}} \nabla_{\theta_{\text{ta}}} \mathcal{J}^e = \theta_{\text{ta}} + \eta_{\text{ta}} \nabla_{\theta_{\text{ta}}} \left( \mathbb{E}_{\pi_{\theta'}}[\sum_{t=0}^{\infty} \gamma^t r_t^e] \right), \tag{4}$$

where $\eta_{\text{ta}}$ is the step size for updating $\theta_{\text{ta}}$. Here, it is noted that the expectation is performed on the updated parameters $\theta'$ that is affected by our intrinsic reward and accordingly $\theta_{\text{ta}}$. Therefore, $\theta_{\text{ta}}$ is updated by the chain rule such that $\nabla_{\theta_{\text{ta}}} \mathcal{J}^e = \nabla_{\theta'} \mathcal{J}^e \nabla_{\theta_{\text{ta}}} \theta'$. This can be considered a well-known bi-level optimization framework popularly used in meta-learning, where the inner-loop and the outer-loop are corresponding to Equation 3 and Equation 4, respectively.

**Model for task-specific modulation.**  After executing an action $a_t$, a task-specific modulation $r_t^{\text{ta}} \in (-1, 1)$ is modeled as

$$r_t^{\text{ta}}(a_{t-1}, s_t, a_t; \theta_{\text{ta}}) = \tanh \left( \mathbb{1}_{a_{t-1}}{}^\top f_{\text{ta}}\big(z_t^e, a_t; \theta_{\text{ta}}\big) \right), \tag{5}$$

where $\mathbb{1}_a$ is the one-hot vector for action $a$, $f_{\text{ta}} : \mathbb{R}^D \to \mathbb{R}^{|\mathcal{A}|}$ is a MLP network, and $z_t^e$ is shared with VQ-VAE based encoding to make the modulator efficient in training. We choose $\tanh$, which generates both positive and negative modulation, so that even in the absence of meaningful signals from episodic counts, the modulator can act according to LIRPG and converge to a value that maximizes the extrinsic objective.

## 4   Experiments

The key concept of LECO is the learnable and modulated episodic count. In other words, compared to a naive episodic count that assigns the state novelty proportional to its visitation count within an episode, LECO aims to produce low intrinsic rewards to task-irrelevant states regardless of its episodic count. We demonstrate such behavior by comparing the mean intrinsic rewards obtained by LECO to those from exploration with just the episodic count-based intrinsic reward in Figure 2. The results show that while the episodic novelty remains high, modulation in LECO decreases the intrinsic reward such that the explorative behavior has vanished towards the end of training and the agent focuses on the task-specific exploitation. Apart from the novelty based on the agent's location, in an environments where agent can pickup and drop objects, such actions can create task-irrelevant novel states, further complicating the search space. The task-specific modulation of LECO can mitigate those task-irrelevant behaviors as shown in Figure 10 and Figure 12 in Appendix.

**Environments.**  As LECO operates on multiple components, we design the experiments to thoroughly cover various aspects, especially the adaptiveness and the scalability. Since the real world as an environment is nonstationary and requires high-dimensional perception, it is crucial for an agent to efficiently explore the constantly changing environment, even at a large scale. Thus, we first demonstrate the explorative performance of LECO in procedurally generated environments of MiniGrid [7]. Then, we further extend the experiments to large scale environments of DMLab tasks.

**Baselines.**  In order to present the contribution of each component of our proposed method, we compare LECO to its variants that incrementally add each component to a baseline without the intrinsic motivation which we denote as *no-Int* for the rest of the section. Similarly, a variant with only the vector quantization episodic counts is denoted as *VQ-only*. Additionally, a variant that uses AE-LSH [33] as the hash function is denoted as *LECO(AE-LSH)*. Furthermore, in order to show that our proposed algorithm is beyond a mere combination of VQ-VAE and LIRPG, we compare the LECO to a straightforward combination of the two, denoted as *LECO-naive*.

The no-Int baseline is an IMPALA-based agent [12], with minor architectural modifications according to the environments. Details on the base model and the architectures can be found in Appendix A. Other baselines include *DSC* (Down Sampled Cell) [10], *AE-LSH* [33], *NovelD* [38], and *AGAC* [13], where DSC uses a non-learnable hash for state compression and AE-LSH uses a learnable hash. NovelD and AGAC are state-of-the-art exploration methods for procedurally generated environments. Here, for a fair assessment of LECO, we implement the episodic count of NovelD and AGAC using the VQ hashing as done in LECO for both of MiniGrid and DMLab tasks.

**Experimental Setup.**  In MiniGrid, LECO was trained using two A100 GPUs with a batch size of 768 for 18 hours. In DMLab, we used eight V100 GPUs with a batch size of 576 for 8 hours. The unroll length was $T = 96$ for all tasks and same LSTM-based policy network architecture was used for LECO and all other baselines. Details on hyperparameters, model architectures, and training

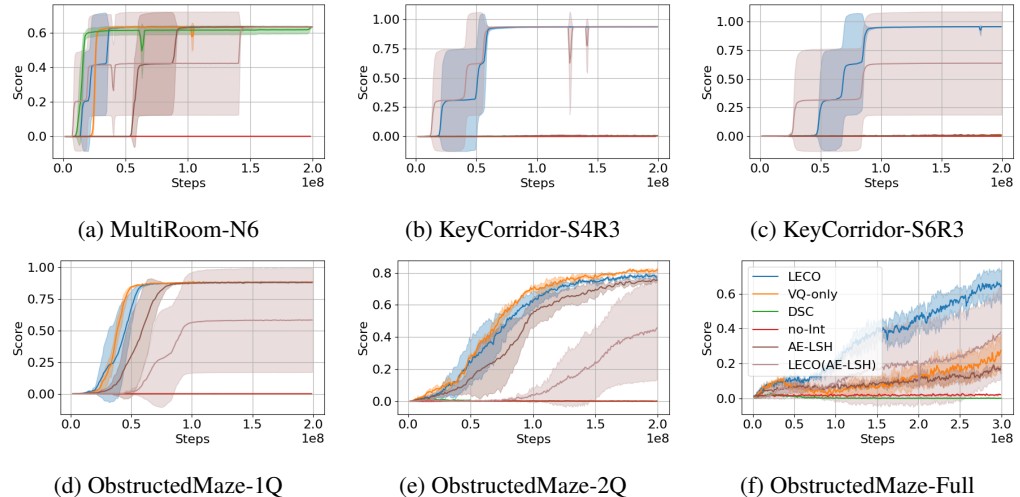

(a) MultiRoom-N6     (b) KeyCorridor-S4R3     (c) KeyCorridor-S6R3

(d) ObstructedMaze-1Q     (e) ObstructedMaze-2Q     (f) ObstructedMaze-Full

Figure 3: Comparison of LECO and its variations on six selected MiniGrid tasks. The final choice of design (LECO) shows superior performance compared to its variants. The shaded area represents a range of standard deviation over 3 runs with different random seeds.

settings are provided in Appendix B. Official codes to run the algorithm and the experiments will be available[2].

## 4.1 MiniGrid

MiniGrid [7] is a procedurally generated gridworld environment specifically designed to test and develop exploration algorithms under sparse rewards and partial observations. A typical world in MiniGrid consists of $M \times N$ tiles, where each tile is assigned to be objects such as floor, wall, lava, door, key, ball, box, and goal. The agent deployed navigates through the world, collecting objects and solving sub-tasks such as opening a door using a key, to reach a goal placed somewhere in the world. Of the many types of environments included, we test the performance of LECO in three types of MultiRoom, KeyCorridor, and ObstructedMaze environments. More specifically, we use total of six configurations of those environments namely `MultiRoom-N6(MRN6)`, `KeyCorridorS4R3(KCS4R3)`, `KeyCorridorS6R3(KCS6R3)`, `ObstructedMaze-1Q(OM1Q)`, `ObstructedMaze-2Q(OM2Q)`, and `ObstructedMaze-Full(OMFull)`, roughly ordered in increasing level of difficulty. The choices were made to reflect a wide range of difficulty including the most difficult task, while evaluating on similar tasks used by the previous baselines.

We emphasize that for all tasks the baselines and LECO only use the original partial input observations, in which a 7x7x3 tensor given as the partially-observable grid cell is the input observation for both the policy network and intrinsic module as in [6]. This setting is more challenging than the other settings such as the partial observation for the policy and full observation for the intrinsic module as in [28, 38, 30], and the partial observation for the policy and meta information such as the agent's position for the intrinsic module as in [13].

We first compare LECO to its variations as shown in Figure 3. The variations include no-Int, DSC, AE-LSH, VQ-only, and LECO(AE-LSH). The results show that LECO solved all six tasks while VQ-only solved three, `MRN6`, `OM1Q`, `OM2Q`, and show slow learning trend in `OMFull`. On the other hand, compared to DSC, learnable hashing methods for using episodic count as an intrinsic reward but with heuristic quantization, VQ-only outperforms DSC as DSC solved only one out of the six tasks. While AE-LSH solved the same number of tasks as VQ-only, the scores of VQ-only are higher in all tasks.

Without the task-specific modulators, the episodic count-only methods such as VQ-only and AE-LSH, tend to perform worse in KeyCorridor tasks than in ObstructedMaze tasks due to the task-irrelevant

[2]https://github.com/kakaobrain/leco

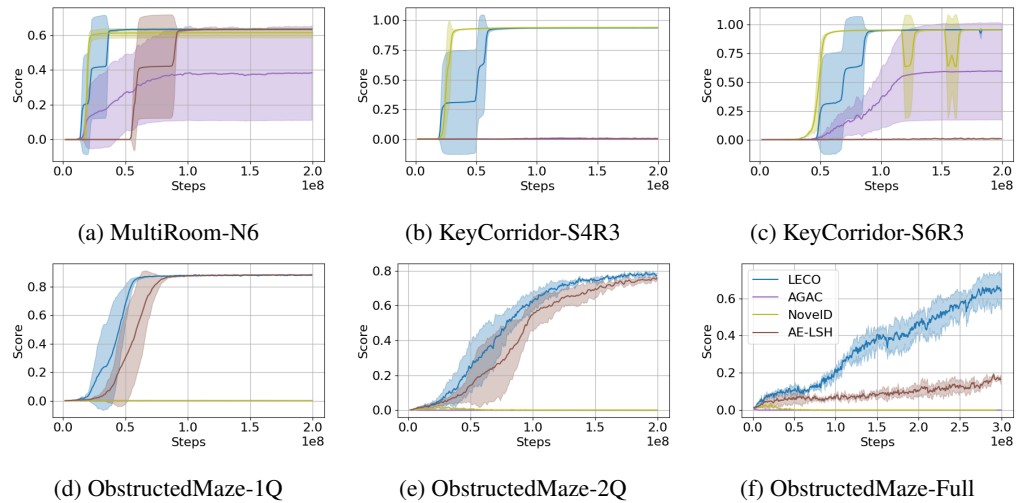

Figure 4: Comparison of LECO to baselines on six selected MiniGrid tasks. LECO shows the most promising performance compared to related existing methods in the field. The shaded area represents a range of standard deviation over 3 runs with different random seeds.

state novelties. More concretely, in `OM1Q` and `OM2Q`, the locked doors are generally located near the keys making it easier for the agent to discover the sequence of opening the locked door by exploration, while in KeyCorridor, even though the map is more simple, the doors are generally placed further away from the keys and thus increased search depth hinders the episodic count-only methods from opening the locked door and eventually solving tasks. We visualize the agent's behavior with and without the modulation in subsection D.1 of Appendix which depicts the episodic count-only agent choosing task-irrelevant actions along the way in KeyCorridor task.

As shown in Figure 3, LECO(AE-LSH) generally performs better than AE-LSH except `OM1Q` and `OM2Q`. However, the performances of LECO(AE-LSH) are lower than those of LECO on all tasks, which shows the benefits from VQ-VAE for the episodic counts. We analyze the benefits of LECO compared to LECO(AE-LSH) over the hash space (see subsection D.3 of Appendix). As shown by the results of no-Int, a policy without any intrinsic reward is unable to solve any of the six tasks.

Combining all comparisons gives indications that 1) episodic count enables training in sparse reward exploration tasks with procedurally generated environments, 2) vector quantization-based counting is more effective than heuristic cell discretization (DSC) as well as AE-LSH, and 3) the task-specific modulation of LECO further improves the performances of the episodic count-based approach.

Moreover, we compare the performance of LECO with other baselines of AE-LSH, AGAC, and NovelD on the same six MiniGrid tasks. As shown in Figure 4, only LECO solves all six tasks including `OMFull`. Here, it is noted that the original implementations of AGAC and NovelD used the agent's position and the full observations, respectively, for the episodic counts in their methods while we used the original partial observations in our implementations of AGAC and NovelD. In fact, the original performances of NovelD and AGAC are better in their papers, and they have also shown to solve OMFull in their papers. However, we observe that, when we correct their official codes to utilize only the partial observations in obtaining the episodic counts, their performances are dropped as in our results. Recently, MADE [37] has shown to produce state-of-the-art performances on MiniGrid including `OMFull`, and MADE seems to be a bit more sample efficient than LECO according to Figure 7 in [37]. However, in this paper, it is somewhat difficult for us to compare the learning progress of LECO with that of MADE since we failed to reproduce their performances given rough descriptions on MiniGrid implementations in their paper and the absence of support for MiniGrid experiments in their official codes. In MultiRoom and KeyCorridor environments, the increasing rates in scores of LECO is slightly slower than those of NovelD since the modulator in LECO requires a bit more experiences to train, however, the advantage of having the modulator is clearly shown through significant performance improvement in the harder exploration tasks. In addition, LECO is more stable in the later phase of KeyCorridor training.

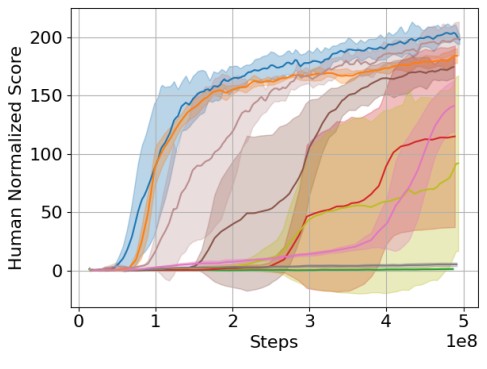 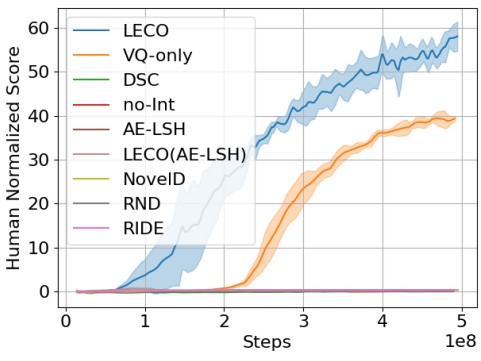

(a) lasertag_three_opponents_small      (b) lasertag_three_opponents_large

Figure 5: Learning curves on two selected DMLab tasks. Compared to all of its variants and baselines LECO shows the highest performance. Especially in `lasertag_three_opponents_large`, only LECO and VQ-only is able to solve the task. The y-axis represents the human normalized score. The shaded area represents the range of a standard deviation over 3 runs of random seeds.

## 4.2 DMLab

DMLab [3] is a learning environment consisting of various tasks, each involving different visuals and objectives in a simulated 3D world. We consider DMLab for its partial-observability that naturally arises from first-person perspective and its complexity from image-based observation with fine details and randomly generated levels to demonstrate the scalability of LECO. We compare LECO and LECO(AE-LSH) to no-Int, episodic count methods based on AE-LSH, DSC, and VQ-only, and state-of-the-art exploration methods including NovelD, RND [5], and RIDE [28] on two individual exploration-heavy tasks. For RND and RIDE, the architectures of neural networks are based on the RND network used in NovelD for MiniGrid and the forward/inverse networks used in RIDE for MiniGrid [3]. We use the same RND networks for RND and NovelD and the same architecture of VQ for LECO, NovelD, and RIDE in their episodic count module.

We select two specific tasks from DMLab, `lasertag_three_opponents_small` and `lasertag_three_opponents_large`, in order to show LECO's capability to solve exploration tasks at a large scale. In both tasks, an agent is deployed in a procedurally generated 3-D environment where it has to search and shoot three opponents moving around the maze-like map (see Figures 16 to 18 for example screenshots). Since the reward is only given when the agent successfully tags the opponents by shooting a laser, these tasks are known to be difficult to solve without an advanced exploration strategy and are characterized as sparse reward problems. The `small` and `large` stand for the size of the map, which determines the reward sparsity. In Figure 5, we present the learning curves of LECO and baselines on these tasks. Each task has its own independent agent, but the architectures and hyperparameters remain consistent across these tasks. To the best of our knowledge, experiments for RND, RIDE, and NovelD on DMLab-lasertags has not been published so far. As shown in Figure 5, these models show decent performances on `lasertag_three_opponents_small`, but perform worse than LECO and moreover do not solve the task of `lasertag_three_opponents_large` at all.

## 4.3 LECO vs LECO-naive.

One of the key contributions of the LECO is the formulation of task-specific modulation of the task-agnostic episodic count in the additive form. In order to show that LECO is more than a simple combination of VQ-VAE and LIRPG, we experimentally compare LECO to a naive combination of VQ-VAE and LIRPG where we directly optimize the episodic counter using the meta-gradient of extrinsic rewards. More concretely, we define the intrinsic reward $r_t^i(a_{t-1}, s_t, a_t, s_{t+1}) = r_t^{ta}(a_{t-1}, s_t, a_t, r_t^{ep}(s_{t+1}))$ and denote as *LECO-naive*. Just from the formulation we can speculate that it would be insufficient to solve sparse reward problems since it

---
[3] `https://github.com/tianjunz/NovelD`

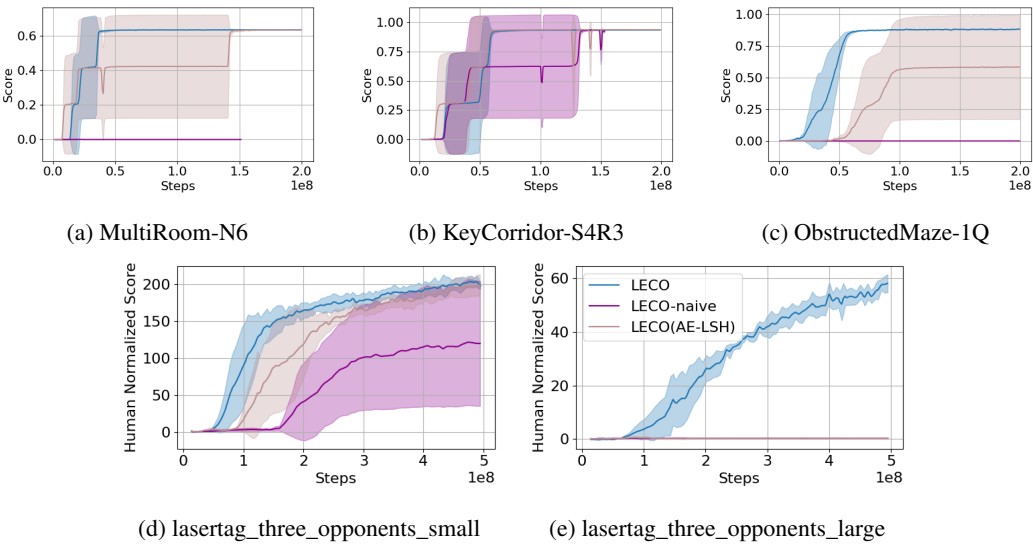

(a) MultiRoom-N6          (b) KeyCorridor-S4R3          (c) ObstructedMaze-1Q

(d) lasertag_three_opponents_small          (e) lasertag_three_opponents_large

Figure 6: Comparison between different variants of LECO on three selected MiniGrid tasks (a)-(c) and the two DMLab tasks (d)-(e). LECO clearly outperforms LECO-naive in all tasks. The shaded area represents a range of standard deviation over 3 runs with different random seeds.

cannot generate dense signals from sparse extrinsic rewards, especially in the early stages of RL. We compare the performances of LECO to LECO-naive along with LECO(AE-LSH) on Minigrid(`MRN6`, `KCS4R3`, and `OM1Q`), and DMLab(`lasertag-three_oppenents_small` and `lasertag-three_oppenents_large`). As shown in Figure 6, the performances of LECO-naive are lower than those of both LECO and LECO(AE-LSH) on all tasks, showing that the proposed additive formulation enables taking the full advantages of both parts and accordingly to solve very hard exploration problems with automatic transition from exploration to exploitation.

## 5   Conclusion

This paper proposes a learnable episodic count to produce task-specific intrinsic rewards for hard exploration problems including high-dimensional state spaces and procedurally generated environments. In specific, the hash code based on VQ-VAE is used for computing the episodic state count and the corresponding state novelty, and this episodic novelty is regulated by the task-specific modulator that is trained to maximize the extrinsic rewards. Experimental results demonstrate that the proposed task-specific exploration based on the episodic count significantly outperforms the previous state-of-the-art exploration methods on the most difficult tasks of MiniGrid and DMLab. However, the proposed intrinsic reward also has a number of limitations. First limitation comes from solving tasks that require intensive leveraging of the inter-episode novelty. For example, objects of negative reward could be common among different episodes, and in that case, the inter-episode novelty can lead an agent to avoid those objects. Second limitation is dealing with somewhat difficult bi-level optimization. The use of implicit gradients and the reformulated single-level problem can be explored to efficiently solve it. For future works, we will extend our work to combine the inter-episode novelty and devise a more efficient training of task-specific intrinsic reward to further improve the scalability. Moreover, we will perform experiments on all DMLab tasks with large-scale transformer-based networks and also apply to the other hard-exploration environments.

## Acknowledgments and Disclosure of Funding

We would like to thank Brain Cloud Team at Kakao Brain for their support.

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
