# OpenReview forum: "LECO: Learnable Episodic Count for Task-Specific Intrinsic Reward"
_NeurIPS.cc/2022/Conference — NeurIPS 2022 Accept_

### Official Review · Reviewer_wRNy · 2022-06-23

**Rating:** 6
**Confidence:** 3
**Soundness:** 3 good
**Presentation:** 3 good
**Contribution:** 2 fair

**Summary:**

This paper studies a learnable hash function for count-based exploration. The hash function is learned through a vector quantized variational auto-encoder. The associated intrinsic reward is further modulated by a second reward that learns to focus on task relevant extrinsic reward, which is learned with a meta-learning approach. Both signals are combined through addition, hoping the task modulation will gradually learn to dominate the novelty, and the agent will gradually switch from exploration to exploitation. Experiments in MiniGrid environments and two DMLab tasks shows that their method generally outperforms or is on par with several baselines.

**Questions:**

-

Conclusion: This is a decent submission: it is a relevant topic, the paper is well written, and their approach outperforms the baseline. My biggest worry is the slightly incremental character of the approach: they essentially glue together two known ideas, which is still a relevant contribution, but not extemely novel. The better performance compared to the baselines is promising, although I slightly miss a description of 1) how baselines were chosen and 2) how test tasks were chosen. Otherwise, this is a good paper.

**Limitations:**

-

**Strengths And Weaknesses:**

Strong:
* Relevant topic (intrinsic motivation).
* Decent experiments with better results than the baselines.
* Well-written paper.

Weak:
* The idea is a bit incremental, since it combines two known ideas: 1) count-based novelty (which was also already tried with VQ-VAE), and 2) meta-optimization of a task-specific intrinsic reward.
* I find it a bit hard to interpret the DMLab results, since there is no figure of the actual task, nor a description how the authors ended up with these two specific tasks. The learning curve of their method does look better.
* I also slightly miss the motivation why these baselines are chosen (DSC, AS-LSH, NovelID, AGAC).

---

> ### Author Response · Authors · 2022-08-02
> **Responses to Reviewer wRNy**
>
> We appreciate your thoughtful review.
>
> **Q1. The idea is a bit incremental, since it combines two known ideas: 1) count-based novelty (which was also already tried with VQ-VAE), and 2) meta-optimization of a task-specific intrinsic reward.**
>
> A: At the surface, our inspirations do originate from those works, however, the core intuition of our proposed method deviates from a mere combination of the two known ideas. Importantly, our contribution lies in recognizing why and how the two ideas can be effectively synthesized. First, the episodic count-based method with VQ-VAE by itself has non-vanishing intrinsic rewards, thus leads an agent to continuously perform meaningless explorations. Simply adding LIRPG on top does not provide any solution to this problem as 1) there is no obvious and straight-forward methodology to directly applying a meta-gradient to count-based algorithms and 2) LIRPG itself requires sufficient extrinsic reward to properly function. Therefore, in order to combine count-based novelty and meta-optimization in a complementary manner, we propose the additive formulation in Eq. 1. Applying the bi-level optimization using this additive form endows the latter part of the addition to directly function as a modulation to control exploration from the episodic counts.
>
> ---
>
> **Q2. I find it a bit hard to interpret the DMLab results, since there is no figure of the actual task, nor a description how the authors ended up with these two specific tasks.**
>
> A: We have added more detailed description of the tasks in the main text (Section 4.2) and some example screenshots in the appendix for visualization. The two tasks, are chosen for three characteristics: 1) procedurally generated high-dimensional input space, 2) sparsity of extrinsic reward and 3) poor performance with non-exploration specific algorithms.
>
> ---
>
> **Q3. I also slightly miss the motivation why these baselines are chosen (DSC, AE-LSH, NovelID, AGAC).**
>
> A: The choice of the baselines is made to reflect a wide spectrum of count-based algorithms that have official codes available and state-of-the-art exploration algorithms that have been successfully applied to MiniGrid tasks.

---

### Official Review · Reviewer_TYQ7 · 2022-06-30

**Rating:** 6
**Confidence:** 4
**Soundness:** 3 good
**Presentation:** 3 good
**Contribution:** 3 good

**Summary:**

This paper introduces a method for count-based intrinsic motivation in Reinforcement Learning. Considering the issue that the counting mechanism may drive the exploration towards states that are known to be useless for maximizing the extrinsic reward, the introduced method LECO modulates the intrinsic reward such that exploration of useless agents is not encouraged. The proposed method uses a neural network for computing the counting (using a variant of a variational autoencoder) and meta-gradient for the training of the modulator. Experiments on meaningful problems are provided.

**Questions:**

- I'm sure I'm missing something, but why the modulation reward is a tanh? In this way, it is unclear to me what the effect on the total reward will be when the modulator reward is > 0. In this way, it seems like the modulator can add an exploratory bonus to some states if they are found to be interesting, is this wanted? It seems more reasonable to me to use a sigmoid and subtract it from the episodic count reward. I'm probably missing something here, so I'd be glad if the authors can clarify this doubt;
- How is the $\lambda$ parameter selected? It seems to me that it should be directly related to the magnitude of the episodic count reward;
- In the appendix, it is stated that the proposed method is similar to LIRPG when the normalized modulation coefficient $\tilde{\lambda} = 1$. Can you clarify what these similarities are?
- Is the need for a vector-quantize VAE crucial? Why not use a regular VAE? Why not a regular AE?

**Limitations:**

The limitations of the proposed work are very briefly explained, from an intuitive point of view, in the conclusion. I think the authors could add more information, for example, describing under which conditions tasks may require intensive leveraging of inter-episode novelty. I also think that the similarity with LIRPG is a limitation that should be addressed properly.

**Strengths And Weaknesses:**

Strengths
-------------
- The paper deals with the interesting field of intrinsic motivation, and studies an important issue affecting methods in literature;
- The proposed method is sound and it seems easy to implement;
- The empirical evaluation provides good evidence of the advantages of the proposed method over several baselines.

Weaknesses
-----------------
- The paper does not provide theoretical results. This is typical for intrinsic motivation method, but it would be nice to see some theoretical guarantees about the convergence to 0 (or at least a very small value) of the intrinsic reward for useless states. Probably this would need a different analysis and implementation allowing closed-form solution, perhaps replacing the MLP of the modulator with a linear approximation;
- It is not clear how the $\lambda$ parameter is computed, although this is not an important weakness and I'm sure it is quite straightforward. Still, I'd like clarifications;
- Most of the equations are inline. This makes it difficult to read sometimes. For example, from line 197 to line 203;
- If the method is similar to LIRPG, why there are no comparisons with it? Am I missing something?

Minor comments
----------------------
- In the list of contributions, the last two items are not contributions; I'd remove them, or rephrase;
- line 121: adapted -> adopted
- Figure 3: visitaion -> visitation

---

> ### Author Response · Authors · 2022-08-02
> **Responses to Reviewer TYQ7 (1/2)**
>
> We appreciate your thoughtful review.
>
> **Q1. The paper does not provide theoretical results. This is typical for intrinsic motivation method, but it would be nice to see some theoretical guarantees about the convergence to 0 (or at least a very small value) of the intrinsic reward for useless states. Probably this would need a different analysis and implementation allowing closed-form solution, perhaps replacing the MLP of the modulator with a linear approximation;**
>
> A: The final range of the intrinsic reward is (-0.5, 1.0) and thus intrinsic reward of 0 is achievable. Under the assumption that we are working in an environment where the intrinsic reward from the episodic count will asymptotically converge to 0, the modulator will act according to LIRPG only and converge to a certain value that maximizes the extrinsic objective.
>
> We include these statements in the revised paper.
>
> ---
>
> **Q2. It is not clear how the λ parameter is computed, although this is not an important weakness and I'm sure it is quite straightforward. Still, I'd like clarifications;**
>
> A: Here, we fix $\lambda$ as a constant hyperparameter and its value is determined by grid search on three MiniGrid tasks as shown in Figure 8 of the appendix.
>
> On the other hand, $\lambda$ can be adaptively calculated relative to the magnitude of the novelty from episodic counts, however, we leave it as the future work.
>
> We include these statements in the revised paper.
>
> ---
>
> **Q3. Most of the equations are inline. This makes it difficult to read sometimes. For example, from line 197 to line 203;**
>
> A: Due to the limitation in space, we had to write several inline equations. We revise the method section to be more clear and self-contained, reducing inline equations.
>
> ---
>
> **Q4. If the method is similar to LIRPG, why there are no comparisons with it? Am I missing something?**
>
> A: The method is similar to LIRPG when $\lambda$ is 1.0 and the performances with $\lambda=1$ are already included in the appendix. As shown in Figure 8, the results show that $\lambda=1$ fails to solve all three tasks.
>
> ---
>
> **Q5. minor comments**
>
> A: The minor typos have been fixed. Thank you for pointing them out.
>
> ---
>
> **Q6. I'm sure I'm missing something, but why the modulation reward is a tanh? In this way, it is unclear to me what the effect on the total reward will be when the modulator reward is > 0. In this way, it seems like the modulator can add an exploratory bonus to some states if they are found to be interesting, is this wanted? It seems more reasonable to me to use a sigmoid and subtract it from the episodic count reward. I'm probably missing something here, so I'd be glad if the authors can clarify this doubt;**
>
> A: Yes, our intention was to give a room for possibly adding more exploratory bonus to some states. The extra bonus may be thought of as a bonus for task-specific exploration and it may also be regarded as an intrinsic motivation to maximize the extrinsic reward, as in LIRPG. Furthermore, we have attempted a similar form to subtracting sigmoid for adaptive modulation but the performance was worse than using tanh.
>
> We include these statements in the revised paper.

---

> ### Author Response · Authors · 2022-08-02
> **Responses to Reviewer TYQ7 (2/2)**
>
> **Q7. How is the λ parameter selected? It seems to me that it should be directly related to the magnitude of the episodic count reward;**
>
> A: While the selection process is directly related to the answer to Q2, we agree on your point that a correlation could be made between lambda and the magnitude of the episodic count. We have attempted few adaptive method for lambda but did not succeed in formulating an effective one. The formulation of such methods is one of our important future works.
>
> ---
>
> **Q8. In the appendix, it is stated that the proposed method is similar to LIRPG when the normalized modulation coefficient λ~=1. Can you clarify what these similarities are?**
>
> A: The task-specific modulation alone in the intrinsic reward of LECO by $\lambda=1$ makes the entire intrinsic reward to be directly optimized by solely extrinsic rewards and more importantly through gradient-based bi-level optimization. This is a similar optimization process of learning the intrinsic rewards in LIRPG. However, the architecture of the task-specific modulator are fairly different from the intrinsic module in LIRPG, and the final optimization loss of the intrinsic module in LECO is also different in that it includes a VQ-VAE loss as a part through the shared encoder.
>
> We will include these statements in the revised paper.
>
> ---
>
> **Q9. Is the need for a vector-quantize VAE crucial? Why not use a regular VAE? Why not a regular AE?**
>
> A: The main purpose of using vector-quantized VAE is to obtain easy state count by mapping massively large state space to a finite discrete space, allowing a fast search based on discrete hash. Such characteristic becomes more crucial especially in the high-dimensional input space and long horizon tasks. This process cannot be done using a regular VAE or AE.
>
> ---
>
> **Q10. I think the authors could add more information, for example, describing under which conditions tasks may require intensive leveraging of inter-episode novelty. I also think that the similarity with LIRPG is a limitation that should be addressed properly.**
>
> A: Tasks will require intensive leveraging of the inter-episode novelty when there are important task-specific information shared across different episodes. For example, meaningless actions could be common among different episodes, and in that case, the inter-episode novelty can lead an agent not to perform those actions.
>
> The key concept of the proposed LECO is the task-specific modulation of the task-agnostic episodic count, as in the additive form of Eq. (1). Our hybrid of the task-agnostic episodic count and the task-specific modulator is different from that directly produces the task-specific intrinsic reward by LIRPG. LIRPG is basically unable to solve very sparse reward problems since it cannot generate dense signals from sparse extrinsic reward, especially in early RL stage.
>
> These statements are included in the revised paper.

---

### Official Review · Reviewer_2TQC · 2022-07-09

**Rating:** 4
**Confidence:** 4
**Soundness:** 2 fair
**Presentation:** 2 fair
**Contribution:** 2 fair

**Summary:**

This paper proposes a new design of intrinsic reward, as a variant of the count-based reward, to tackle the sparse-reward tasks in reinforcement learning.

Prior works related to count-based reward, especially the ones focusing on procedurally generated environments, widely use the episodic count, which means resetting the state visitation count for each episode, instead of recording the global visitation count throughout the entire training with multiple episodes.

This work proposes LECO, i.e. to get the state visitation count via learning the VQ-VAE model and regulate the count-based intrinsic reward via the LIRPG framework. Specifically, VQ-VAE encodes the state into a latent feature and maps the latent feature to the nearest embedding in the code book. The hash code is a sequence of indices of the matched embeddings and the intrinsic reward is based on the visitation count of each hash code in an episode. Also, the count-based reward is combined with the task-specific modulation, which is learned through the LIRPG framework.

The experiment is conducted on procedurally generated environments in MiniGrid benchmark and Deepmind Lab. LEGO is compared with recent works NovelD and AGAC and some variants as ablation study. It turns out that LEGO mostly outperforms these baselines.


**Questions:**

In Section 3, it seems that some details are missed. For example, in Section 3.3, what’s the dimension of $z^e$? What’s the relation between k and $k_i$? Although we can infer these details from the context, it will be better if they are crystal clear. In Equation (2), $log$ is a function, not a probability distribution. What’s the meaning of $log(s_t|z^q)$? Conditional probability distribution? In Section 3.4, $r^i_t$ does not include $\lambda$, which is different from Equation (1). Is it a typo?

Is LECO sensitive to hyper-parameters? such as the size of the codebook in VQ-VAE, the weight of task-specific modulation, the weight of intrinsic reward, etc. Could you please analyze the effect of each critical hyper-parameter? How and why does the higher or lower value choice of each hyper-parameter influence the final performance?

In Figure 5, why does LECO perform worse than NovelD in MultiRoom and KeyCorridor environments, but outperform it in ObstructedMaze? Any intuitive explanation?

In Figure 5, NovelD is worse than the results reported in NovelD paper. Is it because you use VQ-VAE to count instead of using their naive approach to count? Why not use their original setting?

VQ-only in Figure 5 and AE-LSH in Figure 6 shows a weird performance pattern. They can learn reasonably well on easy task MultiRoom and hard task ObstructedMaze, but fail on the medium task KeyCorridor. Why? Have the hyper-parameters (e.g. weight of the intrinsic reward when added upon extrinsic reward) in these two baselines been searched extensively?

In Figure 6, could you explain why LECO is less sample efficient than NovelD for the noisy TV problem? The difference is kind of significant, that is, LECO costs around 4e7 more frames than NovelD to converge.

In Figure 7, it will be more impressive if LECO is compared with more classical and strong methods in Deepmind Lab, such as RND ('Exploration by Random Network Distillation') and ICM (“Curiosity-driven Exploration by Self-supervised Prediction”). NovelD mainly showed advantages in procedurally generated environments, instead of general sparse-reward tasks. Maybe NovelD baseline is not strong enough compared with other possible baselines on Deepmind Lab.

About the ablative study, could you please add the comparison to AE-LASH + LIRPG? I interpreted LECO as VQ-VAE + LIRPG (please correct me if I misunderstand it), so I’m curious about the effect of VQ-VAE in comparison with the widely used counting method AE-LASH.

What has been learned as state features $z^e$? Could you qualitatively or quantitatively analyze it? How about showing some examples of states assigned to the same hash code, and states assigned to different hash codes?

Minor points:
I doubt it is proper to name the memory buffer for hash code counts as ‘episodic memory’ (see Figure 1). In the literature of psychology, this term means recalling particular and subjective life experiences. In the RL area, this term is extended to refer to the state-action pairs with corresponding Q values in a buffer (e.g. ‘Model-Free Episodic Control’, ‘Episodic Memory Deep Q-networks’). The meaning of ‘episodic memory’ in this work seems to deviate from the literature.

The figures look preliminary without axis labels or meaningful sub-captions.


**Limitations:**

The authors mentioned the limitations of LECO on tasks that require inter-episode novelty, or difficult bi-level optimization. The issues are not fully addressed, but it is a natural idea to combine the episodic novelty in LECO with inter-episode novelty.

It will be interesting if the authors study VQ-VAE to count inter-episode state visitation. Especially on hard-exploration tasks on Atari games, the inter-episode novelty is necessary. Could the count learned with VQ-VAE outperform the previous approaches with pseudo-count, e.g. AE-SimHash ('#Exploration: A Study of Count-Based Exploration for Deep Reinforcement Learning'), CoEX ('Contingency-Aware Exploration in Reinforcement Learning'), NGU ('Never Give Up: Learning Directed Exploration Strategies')?


**Strengths And Weaknesses:**

Pros:
It is relatively novel to learn VQ-VAE and then use the discrete hash code from the codebook for state visitation count.
The method is naturally motivated and technically reasonable. Based on prior methods of locality-sensitive hashing with auto-encoder (VE), it is a natural idea to modify AE to VAE. Because we need discrete state representation for counting, it is natural to try VQ-VAE.
The experiment result is positive, especially on hard exploration tasks ObstructedMaze in MiniGrid.

Cons:
The novelty is a bit limited. LECO seems simply a combination of two prior works: VQ-VAE and LIRPG. It will be great if you can clarify more technical contributions beyond the combination.
The method is not clearly explained in Section 3.3 and 3.4. Ideally, the method section should be self-contained enough for readers, even though they are not familiar with VQ-VAE or LIRPG.
The experiment part to compare with baselines can be improved.  The comparison with the most recent SOTA on MiniGrid (MADE) is missed. The performance of baseline NovelD in Figure 5 is worse than the results reported in NovelD paper.

---

> ### Author Response · Authors · 2022-08-02
> **Responses to Reviewer 2TQC (1/3)**
>
> We appreciate your thoughtful review.
>
> **Q1. The novelty is a bit limited. LECO seems simply a combination of two prior works: VQ-VAE and LIRPG. It will be great if you can clarify more technical contributions beyond the combination.**
>
> A: The key concept of the proposed LECO is the task-specific modulation of the task-agnostic episodic count, as in the additive form of Eq. (1).
> Of course, the use of VQ-VAE itself for scalable and fast episodic counting in a high-dimensional space and long episode time has not been thoroughly used in the existing exploration methods. On top of that, we propose to apply a task-specific modulator which additively scales the task-agnostic state novelty.
>
> Our hybrid of the task-agnostic episodic count and the task-specific modulator is not a simple combination of VQ-VAE and LIRPG. If we directly optimize the episodic counter using the meta-gradient of extrinsic rewards like LIRPG, (we observe that) it would be unable to solve sparse reward problems since it cannot generate dense signals from sparse extrinsic rewards, especially in early RL stage. Our additive formulation enables taking the full advantages of both parts and accordingly to solve very hard exploration problems with automatic transition from exploration to exploitation.
>
> We clarify our key concept and technical contributions in the revised paper.
>
> ---
>
> **Q2. The method is not clearly explained in Section 3.3 and 3.4. Ideally, the method section should be self-contained enough for readers, even though they are not familiar with VQ-VAE or LIRPG.**
>
> A: Due to the space limitation, we chose to be a little concise in this part. Taking your advice, we revise the method description to be more clear and self-contained. Please check the revised paper.
>
> ---
>
> **Q3. The experiment part to compare with baselines can be improved. The comparison with the most recent SOTA on MiniGrid (MADE) is missed. The performance of baseline NovelD in Figure 5 is worse than the results reported in NovelD paper.**
>
> A: First, we failed to reproduce the results of MADE on MiniGrid. Our own implementation of MADE did not solve any MiniGrid task. This could be due to the wrong choices of hyperparameters missing details on the implementation since there is no detailed description on their implementation and their choice of hyperparameter values, even though these values would be important in obtaining their results. Moreover, their official codes do not support experiments on MiniGrid. We think that MADE cannot be fairly compared without more details on their experiments on MiniGrid.
>
> Regarding NovelD, ***we already noted in our paper that the original NovelD makes use of the full observations when computing its episodic counts, which we confirmed from their official codes. In fact, the MiniGrid tasks allow using only the original partial input observations, and we follow this original, more challenging, and more fair setting for all compared methods including our LECO in our paper***. We also observe that when we modify their official codes to utilize the partial observations in obtaining the episodic counts used in NovelD, their performances are substantially dropped on difficult tasks like the results we reported in our paper.
>
> We include these statement in the revised paper.
>
> ---
>
> **Q4.  In Section 3, it seems that some details are missed. For example, in Section 3.3, what’s the dimension of ze? What’s the relation between k and ki? Although we can infer these details from the context, it will be better if they are crystal clear. In Equation (2), log is a function, not a probability distribution. What’s the meaning of log(st|zq)? Conditional probability distribution? In Section 3.4, rti does not include λ, which is different from Equation (1). Is it a typo?**
>
> A: Please refer to appendix for detailed choices of the hyperparameters. Specifically, as shown in Table 2 in the appendix, the dimension of $z^e$ is 64.
>
> As of $k$ and $k_i$, $k$ is the code index in the codebook applying on a certain spatial position whereas $k_i$ is the code index of the ith spatial position. We revise Section 3.3 to make these notations to be more clear.
>
> As of $log(s_t \vert z^q)$ and the missing lambda in Eq. (1), they are indeed typos. We thank you for pointing them out. $\log(s_t \vert z^q)$ is properly changed to $\log p(s_t \vert z^q)$ and missing lambda is added in the revised paper.

---

> > ### Comment · Reviewer_2TQC · 2022-08-08
> > **Thank Authors for the Detailed Response**
> >
> > I appreciate the detailed response to each of my questions. I could see some improvement in the presentation and analysis of the paper in the updated version. Especially, I found the analysis of the learned hash code interesting. So I raise my rating a bit.
> >
> > About the technical novelty, as you explained, "If we directly optimize the episodic counter using the meta-gradient of extrinsic rewards like LIRPG, (we observe that) it would be unable to solve sparse reward problems since it cannot generate dense signals from sparse extrinsic rewards, especially in early RL stage". It would be more convincing if the experiment of the naive combination is included in the paper.
> >
> > The remaining concern lies in the evaluation. I understand that the author changed NovelD for comparison and met trouble in replicating MADE. However, I'm unsure whether it is a good practice to change/discard previous SOTA by claiming they have issues in implementations/reproducibility. And at the same time, the code for LECO itself is not provided at all, so its own reproducibility can not be guaranteed either.

---

> > > ### Author Response · Authors · 2022-08-09
> > > **We really appreciate your kind comments again.**
> > >
> > > - Experiments of Naive Combination
> > >     - We experimentally compare LECO to a naive combination of VQ-VAE and LIRPG where we directly optimize the episodic counter using the meta-gradient of extrinsic rewards. More concretely, we define the intrinsic reward $r^i_t(a_{t-1}, s_t, a_t, s_{t+1})$ =  $r_t^{ta}(a_{t-1}, s_{t}, a_t, r_t^{ep}(s_{t+1}))$ and it is denoted as *LECO-naive*.
> > >     - We experiment LECO-naive on Minigrid(MRN6, KCS4R3, and OM1Q), and DMLab(lasertag-three\_oppenents\_small and lasertag-three\_oppenents\_large). The results are shown in Figure 11 in Appendix.
> > >     - As shown in the results, the performances of LECO-naive are lower than those of LECO on all tasks, which shows that the proposed additive formulation enables taking the full advantages of both parts and accordingly to solve very hard exploration problems, especially with very sparse rewards in the early RL stage, by automatic transition from exploration to exploitation.
> > >
> > >      We include this in the revised paper (see subsection.5 of Appendix.D).
> > >
> > >
> > > - Comparison with NovelD and MADE
> > >     - First, we would like to point out that NovelD does not clearly state the use of full observations in computing the episodic counts in the paper; we observe it from their codes and also verify that such feature is critical for NovelD in solving the hard MiniGrid tasks. In our paper, we point this out and compare it to LECO under more strict and fair setting with only partial observations. We do not reduce their performances on purpose without any comment in our paper. We think that our comparison under fully partial observation is more appropriate.
> > >     - Second, in our paper, we **clearly state** that MADE has also shown to solve OMFull in their paper. But, we cannot compare the learning curves of MADE with LECO since we failed to reproduce the reported performances (which was far from close) and their official codes do not support MiniGrid experiments. In fact, the first author of NovelD is same with that of MADE, and we could not understand why the codes related to MiniGrid experiments were left out in the official repository of MADE while it is included in the official codes of NovelD.
> > >     - To this end, reflecting your advice, we revise our paper to **remove the statement on the first success in solving OMFull by LECO** and to **include the comments on the original performances of NovelD and MADE** as follows:
> > >     In fact, the original performances of NovelD and AGAC are better in their papers, and they have also shown to solve OMFull in their papers. However, we observe that, when we correct their official codes to utilize only the partial observations in obtaining the episodic counts, their performances are dropped as in our results. Recently, MADE [35] has shown to produce state-of-the-art performances on MiniGrid including OMFull, and MADE seems to be a bit more sample efficient than LECO according to Figure 7 in [35]. However, in this paper, it is somewhat difficult for us to compare the learning progress of LECO with that of MADE since we failed to reproduce their performances given rough descriptions on MiniGrid implementations in their paper and the absence of support for MiniGrid experiments in their official codes.
> > >
> > >
> > > - LECO codes
> > >     - We upload our LECO codes in the supplementary material. You can reproduce our results by running commands in Readme.

---

> ### Author Response · Authors · 2022-08-02
> **Responses to Reviewer 2TQC (2/3)**
>
> **Q5. Is LECO sensitive to hyper-parameters? such as the size of the codebook in VQ-VAE, the weight of task-specific modulation, the weight of intrinsic reward, etc. Could you please analyze the effect of each critical hyper-parameter? How and why does the higher or lower value choice of each hyper-parameter influence the final performance?**
>
> A: The ablation on the codebook size of VQ-VAE and the weight of task-specific modulation $\lambda$ are already included in the Appendix.C.
>
> Regarding the codebook size, for Minigrid, we test the hash sizes of {3x3x8, 2x2x8, 3x3x16, 4x4x8} while for DMLab we test the hash sizes of {6x4x24, 6x4x12, 3x2x36, 3x2x24, 3x2x12}. Here, the hash size is the product of the spatial size ($w \times h$) and the codebook size ($K$). As shown in Figure 7, the performances of LECO would be somewhat sensitive to the hash size. The spatial size would be a little more influential than the codebook size. Here, it is noted that we fix the hash size for all tasks on each benchmark.
>
> In addition, the performances of LECO according to the weight of task-specific modulation are shown in Figure 8. Either the episodic state novelty (weight=0) or the task-specific modulation (weight=1) results in failure in solving the task. On the other hand, putting more weight on the episodic state novelty (weight=0.25) relatively performs better than small weight (weight=0.75), however the performance is best when the two terms are balanced (weight=0.5).
>
> ---
>
> **Q6. In Figure 5, why does LECO perform worse than NovelD in MultiRoom and KeyCorridor environments, but outperform it in ObstructedMaze? Any intuitive explanation?**
>
> A: In MultiRoom and KeyCorridor environments, the scores of LECO increase a bit slower that those of NovelD since the modulator in LECO requires a bit more experiences. However, the advantage of having the modulator shows clear improvement in the harder exploration tasks. In addition, LECO is more stable in the later phase of KeyCorridor training.
>
> We include these statements in the revised paper.
>
> ---
>
> **Q7. VQ-only in Figure 5 and AE-LSH in Figure 6 shows a weird performance pattern. They can learn reasonably well on easy task MultiRoom and hard task ObstructedMaze, but fail on the medium task KeyCorridor. Why? Have the hyper-parameters (e.g. weight of the intrinsic reward when added upon extrinsic reward) in these two baselines been searched extensively?**
>
> A: First,  we observe that the change of alpha (weight of the intrinsic reward when added upon extrinsic reward) do not improve the performances of VQ-only and AE-LSH on KC tasks.
>
> For the count-only methods (i.e. VQ-only and AE-LSH), KC tasks seem to be harder than OM tasks due to the task-irrelevant state novelties. More concretely, in OM1Q and OM2Q, the locked door is often located near the keys and therefore the agent can learn to open the locked door readily while in KC, even though the map is more simple, the door is generally further away from the key and task-irrelevant actions and the corresponding high state novelties hinder the count-only based methods from opening the locked door and eventually solving tasks.
>
> We include these statements in the revised paper.
>
> ---
>
> **Q8. In Figure 6, could you explain why LECO is less sample efficient than NovelD for the noisy TV problem? The difference is kind of significant, that is, LECO costs around 4e7 more frames than NovelD to converge.**
>
> A: We conjecture that the task-specific modulator in LECO requires a little more experiences than NovelD since it is updated upon the extrinsic rewards that are sparse in the early RL phase.  In a similar argument, DSC, which requires no training, is most efficient in the noisy TV problem. Moreover, in fact NovelD converges stably in about 3e7 frames (vs. 5e7 frames for LECO).
>
> The noisy TV experiment has been moved to the appendix due to the space limitation in the main text. The explanation above is added to the respective section of the appendix.

---

> ### Author Response · Authors · 2022-08-02
> **Responses to Reviewer 2TQC (3/3)**
>
> **Q9. In Figure 7, it will be more impressive if LECO is compared with more classical and strong methods in Deepmind Lab, such as RND and ICM.**
>
> A: Following your advice, we also perform RND and RIDE (an advanced version of ICM for procedurally generated environment) on the DMLab tasks. For RND and RIDE, the architectures of neural networks are based on the RND network used in NovelD for MiniGrid and the forward/inverse networks used in RIDE for MiniGrid. The episodic count in RIDE is implemented by VQ-VAE as in NovelD and LECO. For RND, RIDE, and NovelD, we use the learning rate of 0.0001 and the intrinsic reward coefficient of 0.01 which were the best values found for these methods. Their learning curves are added to Figure 5 of the revised paper.
>
> To the best of our knowledge, experiments for RND, RIDE, and NovelD on DMLab-lasertags has not been published so far. As shown in Figure 5, these models show decent performances on lasertag_three_opponents_small, but perform worse than LECO and moreover do not solve the task of lasertag_three_opponents_large at all.
>
> ---
>
> **Q10. About the ablative study, could you please add the comparison to AE-LSH + LIRPG? I interpreted LECO as VQ-VAE + LIRPG (please correct me if I misunderstand it), so I’m curious about the effect of VQ-VAE in comparison with the widely used counting method AE-LSH.**
>
> A: In LECO, any count-based method can be used in obtaining $r^\text{ep}$. Hence, AE-LSH+LIRPG you mentioned can be a variant of LECO.
>
> Here, please note that LECO is not a simple combination of VQ-VAE + LIRPG as the response of Q1. Our hybrid of the task-agnostic episodic count and the task-specific modulator is different from that directly produces the task-specific intrinsic reward by LIRPG.
>
> Having said that we perform experiments of this variant (denoted as LECO(AE-LSH)) on both MiniGrid and DMLab. As shown in Figure 3 and 5, LECO(AE-LSH) generally performs better than AE-LSH except OM1Q and OM2Q. However, the performances of LECO(AE-LSH) are lower than those of LECO on all tasks, which shows the benefits from VQ-VAE for the episodic counts.
>
> We include these statements in the revised paper.
>
> ---
>
> **Q11. What has been learned as state features ze? Could you qualitatively or quantitatively analyze it? How about showing some examples of states assigned to the same hash code, and states assigned to different hash codes?**
>
> A: Following your advice, we analyze the hashing results obtained by VQ-VAE, especially compared to those from AE-LSH.
>
> First, we quantitatively show the effectiveness of VQ compared to AE-LSH in Table 5 of the Appendix in the revised paper. On the DMLab-Lasertag_three_opponents the new hash rates, which measure how often the states are mapped into a new hash within an episode, obtained by VQ are smaller than those by AE-LSH even though the entire hash space of VQ is larger than AE-LSH (i.e. $6^{24} > 2^{62}$). This means that VQ makes similar states to be more grouped into the same hash.
>
> We qualitatively show some weaknesses of AE-LSH compared to VQ in the Appendix.D of the revised paper. In general, both hash methods map similar states into the same hash, however, some differences can be observed in the obtained hashing results (see Figures 13-17 of the Appendix).
>
> - As shown in Figure 13 and 14, in Minigrid, AE-LSH maps important states (i.e. states in which the agent realizes a target object or states just before obtaining a target object) and relatively less important states into the same hash. This obstructs correctly distinguishing task-specific state novelty for efficient exploration. On the other hand, VQ more clearly clusters states to the hash according to the relative importance.
> - In DMLab-lasertags, the difference in performance between AE-LSH and VQ is also reflected in the hashing results. As shown in Figures 15-17, AE-LSH maps states to hashes according to visual similarities whereas VQ learns to map according to the importance represented via the presence of the opponents in sight.
>
> ---
>
> **Q12. I doubt it is proper to name the memory buffer for hash code counts as ‘episodic memory’ (see Figure 1). In the literature of psychology, ....**
>
> A: Our initial intention was to be consistent with the usage of the term episodic memory in NGU. However, according to your suggestion, we rename it as ‘**episodic hash memory**’ in the revised paper.
>
> ---
>
> **Q13. The figures look preliminary without axis labels or meaningful sub-captions.**
>
> A: The figures are fixed accordingly in the revised paper. Specifically, the axis labels are added and the messages of figures are updated in sub-captions.
>
> ---
>
> **Q14. It will be interesting if the authors study VQ-VAE to count inter-episode state visitation. Especially on hard-exploration tasks on Atari games, the inter-episode novelty is necessary.**
>
> A: That is a very intriguing point. We will further investigate into this as part of the future direction.

---

### Meta-Review · Area_Chair_pqZN · 2022-08-27

**Recommendation:** Accept
**Confidence:** Less certain

**Metareview:**

Reviews were mixed here and all quite borderline. There are legitimate points raised for why this is being consistently given borderline ratings, with two in particular resonating with my own reading (novelty and comparisons with other methods). However, despite these issues the paper itself is a solid contribution, and I think could easily lead to others building on the core ideas and approach. The paper has also improved notably during revisions and at present does not have any fundamental flaws that should preclude publication. Therefore, I recommend acceptance.

**Award:**

No

---

### Decision · Program_Chairs · 2022-09-14

Accept